# A Multistage Formulation Based on Full-Length CSP and AMA-1 Ectodomain of *Plasmodium vivax* Induces High Antibody Titers and T-cells and Partially Protects Mice Challenged with a Transgenic *Plasmodium berghei* Parasite

**DOI:** 10.3390/microorganisms8060916

**Published:** 2020-06-17

**Authors:** Luciana C. Lima, Rodolfo F. Marques, Alba Marina Gimenez, Katia S. Françoso, Eduardo Aliprandini, Tarsila M. Camargo, Anna Caroline C. Aguiar, Dhelio B. Pereira, Laurent Renia, Rogerio Amino, Irene S. Soares

**Affiliations:** 1Department of Clinical and Toxicological Analyses, School of Pharmaceutical Sciences, University of São Paulo, São Paulo, SP 05508-000, Brazil; dpandora@gmail.com (L.C.L.); rodolfoferreira@usp.br (R.F.M.); albamarinagimenez@gmail.com (A.M.G.); katiafrancoso@usp.br (K.S.F.); tarsilacamargo82@gmail.com (T.M.C.); 2São Carlos Institute of Physics, University of São Paulo, São Carlos, SP 13563-120, Brazil; carolcaguiar@yahoo.com.br; 3Centro de Pesquisas em Medicina Tropical, Porto Velho, RO 76812-329, Brazil; dbpfall@gmail.com; 4Singapore Immunology Network, Biopolis, Agency for Science Technology and Research, Singapore 138632, Singapore; renia_laurent@immunol.a-star.edu.sg; 5Unit of Malaria Infection & Immunity, Institut Pasteur, 75015 Paris, France; eduardo.aliprandini@pasteur.fr (E.A.); rogerio.amino@pasteur.fr (R.A.)

**Keywords:** malaria vaccine, *Plasmodium vivax*, circumsporozoite protein, apical membrane antigen 1

## Abstract

Infections with *Plasmodium vivax* are predominant in the Americas, representing 75% of malaria cases. Previously perceived as benign, malaria vivax is, in fact, a highly debilitating and economically important disease. Considering the high complexity of the malaria parasite life cycle, it has been hypothesized that an effective vaccine formulation against *Plasmodium* should contain multiple antigens expressed in different parasite stages. Based on that, we analyzed a recombinant *P. vivax* vaccine formulation mixing the apical membrane antigen 1 ectodomain (PvAMA-1) and a full-length circumsporozoite protein (PvCSP-All_FL_) previously studied by our group, which elicits a potent antibody response in mice. Genetically distinct strains of mice (C57BL/6 and BALB/c) were immunized with the proteins, alone or in combination, in the presence of poly(I:C) adjuvant, a TLR3 agonist. In C57BL/6, high-antibody titers were induced against PvAMA-1 and the three PvCSP variants (VK210, VK247, and *P. vivax*-like). Meanwhile, mixing PvAMA-1 with PvCSP-All_FL_ had no impact on total IgG antibody titers, which were long-lasting. Moreover, antibodies from immunized mice recognized VK210 sporozoites and blood-stage parasites by immunofluorescence assay. However, in the BALB/c model, the antibody response against PvCSP-All_FL_ was relatively low. PvAMA-1-specific CD3^+^CD4^+^ and CD3^+^CD8^+^ T-cell responses were observed in C57BL/6 mice, and the cellular response was impaired by PvCSP-All_FL_ combination. More relevant, the multistage vaccine formulation provided partial protection in mice challenged with a transgenic *Plasmodium berghei* sporozoite expressing the homologous PvCSP protein.

## 1. Introduction

Globally, there were an estimated 228 million malaria cases and 405,000 malaria deaths in 2018 according to the World Health Organization (WHO) [1]. Vaccines have been responsible for the control, prevention, and eradication of many infectious diseases. However, the development of vaccines targeting parasites, such as those targeting *Plasmodium* protozoans, the causative agent of malaria, is very complex [2]. Clinical trials have focused almost entirely on *Plasmodium falciparum*, and the most advanced malaria vaccine candidate is RTS,S based on the circumsporozoite protein (CSP), now named Mosquirix™. This vaccine has been evaluated in a large Phase 3 trial [3] and was recently (April 2019) recommended by WHO for large-scale pilot implementations in areas of moderate-to-high malaria transmission in Africa [4]. By contrast, clinical trials with *P. vivax* have been neglected, with only two studies reported (reviewed in [5]).

Considering the promising RTS,S results [4], our group [6,7,8] and others (reviewed in [9]) have invested in the CSP as a target for *P. vivax* vaccines. However, keeping in mind the highly complex life cycle and genetic *Plasmodium* variability, it has been hypothesized that a multiantigen and multistage formulation would be more effective [10]. In this context, additional target antigens, including *P. falciparum* apical membrane antigen 1 (PfAMA-1), have been tested in clinical studies combined with PfCSP [11,12,13,14].

*P. vivax* CSP vaccine has also been combined into multivalent formulations or chimeric synthetic molecules. Peptides based on the regions N-terminal, central repeats, and C-terminal of PvCSP were immunogenic in individual administrations of BALB/c mice [15], *Aotus* monkeys [16], and healthy human volunteers [17]. The most advanced recombinant protein formulation for *P. vivax*, VMP001, merges in central region variant epitopes VK210 and VK247 and was proven immunogenic in mice [18,19,20,21], *Rhesus* monkeys [20,22], and human naive volunteers [23]. However, these vaccines did not consider the three allelic variants of *P. vivax* CSP (VK210, VK247, and *P. vivax*-like) that differ in terms of the amino acid sequence of the central protein region, which has B-cell epitopes [24,25,26].

Our group demonstrated high immunogenicity with constructs fusing all three PvCSP variants (VK210, VK247, and *P. vivax*-like) in C57BL/6 mice [6,7,8]. These recombinant multivariant chimeric recombinant proteins were expressed in *Pichia pastoris* yeast and comprised (i) the conserved region I (RI), which is reported to be a target for protective antibodies [27,28], followed by an immunodominant central repeat domain representing the three variant repeats in tandem and the C-terminal domain (PvCSP-All_CT_) and (ii) a second recombinant protein, named PvCSP-All_FL_, containing the complete N-terminal domain, including RI region, the central repeat domain, and the C-terminal domain. More relevant, both constructs formulated in the presence of the TLR3 agonist poly(I:C) conferred partial protection in models of murine malaria against Pb/Pv sporozoite (i.v.) challenge [8], though only PvCSP-All_CT_ efficacy was tested against Pb/Pv sporozoite (s.c.) challenge [7]. Previous studies with PfCSP have demonstrated that the antibody response to the N-terminal is associated with protection [29,30].

One of the most studied and well-characterized *P. falciparum* blood-stage antigens for the purpose of composing a vaccine against malaria is AMA-1, with different formulations being assessed and tested in malaria-endemic areas in Africa [31,32,33,34]. On the other hand, little is known about the immune response induced by AMA-1 of *P. vivax*. Our group has shown that recombinant constructions of PvAMA-1 are recognized by antibodies induced by natural infection [35,36,37,38,39] and experimental immunizations [38,39,40]. The recombinant protein PvAMA-1 produced in *Pichia pastoris* in the presence of the adjuvant QuilA (saponin isolated from the bark of the *Quillaja saponaria* tree) inhibited the reticulocyte invasion of four different *P. vivax* Thailand isolates [39]. These promising results justified its inclusion in this work to obtain a multistage vaccine formulation. This work describes the immunogenicity analysis of vaccine formulations composed only of the chimeric PvCSP-All_FL_ and the influence of PvAMA-1 combination.

## 2. Materials and Methods

### 2.1. Recombinant Protein Expressed in Pichia Pastoris

The recombinant protein PvCSP-All_FL_ has been recently described [8]. This protein contains the N- and C-terminal regions and the central repeats sequence of *P. vivax* allelic variants (Figure 1). The central region contains six copies of the VK210 sequence (GDRA[A/D]GQPA), followed by six copies of *P. vivax*-like repeats (APGANQEGGAA) and five copies of the VK247 sequence (ANGAGNQPG). The PvAMA-1 recombinant protein has also been previously described [39]. Briefly, selected *P. pastoris* clones were grown for 24 h at 28–30 °C, 230 rpm, in 1 L of BMGY (1% [wt/vol] yeast extract [Sigma-Aldrich, St. Louis, MO, USA], 2% [wt/vol] peptone [Sigma-Aldrich, St. Louis, MO, USA], 1.34% [wt/vol] yeast nitrogen base without amino acids [Sigma-Aldrich, St. Louis, MO, USA], 4 × 10^−5^ % [wt/vol] biotin [Sigma-Aldrich, St. Louis, MO, USA], 1% [wt/vol] glycerol [Sigma-Aldrich, St. Louis, MO, USA], 0.1 M potassium phosphate [Sigma-Aldrich, St. Louis, MO, USA] [pH 6.0]) medium. After this period, cells were harvested by centrifugation, resuspended in 200 mL of BMMY (BMGY with glycerol replaced by 0.5% [vol/vol] methanol [Merck, Darmstandt, Germany]) medium, and cultured for an additional period of 72 h. The induction was maintained by methanol (Merck Millipore, Billerica, MA, USA) addition to a final concentration of 1%. The recombinant proteins were purified by affinity using a HisTrap™ FF nickel column and ion exchange using a QFF HiTrap™ column (GE Healthcare USA Inc., Pittsburgh, PA, USA) chromatography coupled to an ÄKTA prime plus system (GE Healthcare USA Inc., Pittsburgh, PA, USA). Selected fractions were dialyzed against phosphate-buffered saline (PBS, pH = 7.4) and quantified by prediction analysis using ImageQuant^®^ software (GE Healthcare USA Inc., Pittsburgh, PA, USA) compared to defined concentrations of bovine serum albumin (BSA) (Sigma-Aldrich, St. Louis, MO, USA).

### 2.2. Recombinant Proteins Expressed in Escherichia coli

Details of the generation of the three constructs with N- and C- terminal portions and the three FliC-PvCSP repeats (FliC-PvCSP-VK210, FliC-PvCSP-VK247, and FliC-PvCSP-*P. vivax*-like) used in specificity antibody analysis have been previously described by our group [37,39,41,42]. The recombinant proteins were checked by 12% SDS-PAGE stained with Coomassie blue and analyzed by immunoblotting using a monoclonal anti-His tag (1:1000), anti-PvCSP-VK210 (1:200), and anti-PvCSP-VK247 (1:200 as previously described) [7].

### 2.3. Animals

Female BALB/c (H-2^d^) or C57BL/6 (H-2^b^) mice aged 6–8 weeks were immunized with 10 µg of each recombinant protein, individually or as a multiantigen mix, in the presence of 50 μg/dose of polyinosinic–polycytidylic acid adjuvant, poly(I:C) HMW (Invivogen, San Diego, CA, USA). A total of three subcutaneous (s.c.) immunizations (100 µL) were administered with an interval of 15 days. Controls received only adjuvant diluted in PBS. Serum samples were collected for analysis 14 days after each dose and stored at −20 °C. For longevity analysis, mice were followed for 420 days after the first dose, and sera were collected every 30 days. All animal experiments were approved by the Animal Care and Use Committee of the University of São Paulo (CEUA/FCF 362/2012).

### 2.4. Analysis of IgG Antibody by ELISA

Antibodies against PvCSP-All_FL_ and PvAMA-1 in mice sera were detected by ELISA on days 14, 29, and 44 as described previously [7,8,39]. Briefly, high-binding 3590 Costar plates (Corning, New York, NY, USA) were coated overnight at room temperature (RT) with 200 and 100 ng/well, respectively, of each homologous recombinant protein. Plates were washed with PBS-T, blocked 2 h at 37 °C with PBS–milk–BSA (PBS, pH = 7.4, containing 5% nonfat dry milk (Molico^®^, Nestlé S.A., Vevey, VD, Switzerland) and 2.5% BSA), and incubated with serial dilutions of serum from immunized mice, with dilutions beginning with 1:100, for 1 h at RT. Plates were then washed three times with PBS-T, and a solution containing peroxidase-conjugated goat antimouse IgG diluted (1:3000) (Sigma-Aldrich, St. Louis, MO, USA) was added to each well. Detection of IgG subclass responses was performed as described above, except that the secondary antibody was specific to mouse IgG1, IgG2a, IgG2b, IgG2c, or IgG3 (Southern Technologies Birmingham, AL, USA) diluted 1:8000. Because the IgG2a gene is deleted in C57BL/6, we measured IgG2c in these animals. The specific titers were determined as the highest dilution yielding an OD_492_ greater than 0.1. The results are expressed as means of IgG titers (log_10_) ± SEM.

### 2.5. Indirect Immunofluorescence Assay

Thin-smear preparations containing the *P. vivax* sporozoite VK210 subtype were obtained from mosquitoes fed on infected patient as previously described [7,8], and merozoite from *P. vivax* were obtained from malaria-infected patients as described. The *P. vivax*-infected blood from patients attending the Centre of Malaria Control (CEPEM) in the city of Porto Velho, state of Rondônia, in the Brazilian Western Amazon, was collected after written informed consent was obtained (Ethics Committee from the Centro de Pesquisa em Medicina Tropical (CEPEM), Rondônia, CAAE 61442416.7.0000.0011).

In this study, only *P. vivax* monoinfected patients with parasitaemia between 2000 and 80,000 parasites/μL were recruited. From each volunteer, a peripheral venous blood sample (5 mL) was collected by venipuncture in heparin-containing tubes and immediately performed as described. White blood cells and platelets were removed using a CF11 column [43]. The *P. vivax*-infected erythrocytes were cultured to the late schizont stage in 2% hematocrit using McCoy’s 5A medium (Sigma-Aldrich, St. Louis, MO, USA) supplemented with 2.4 g/L D-glucose (Sigma-Aldrich, St. Louis, MO, USA), 40 mg/mL gentamicin sulfate, and 20% heat-inactivated human AB serum in an atmosphere of 5% O_2_ at 37.5 °C. The parasite culture was monitored until they reached at least 40% of the morphology in the mature schizont stage. The mature schizonts were concentrated on a cushion of 45% Percoll (Sigma-Aldrich, St. Louis, MO, USA) centrifuged for 15 min at 1600 G [44]. After being washed twice in McCoy’s 5A medium (Sigma-Aldrich, St. Louis, MO, USA), thin-smear preparations of the schizont concentrate were smeared onto glass slides, air-dried, and fixed with cold acetone for 15 min, then stored at 20 °C until needed.

The slides were blocked with 5% BSA in PBS solution for 60 min at 37 °C. Sera from C57BL/6 mice immunized as described above (1:100 dilution) were applied to the slides and incubated for 1 h RT. MAb anti-K243 DII-AMA-1 (1:1000) and mAb anti-CSP-VK210 (1:100) were used as positive controls. Washing of the slides was done three times in PBS prior to the addition of Alexa Fluor 568 (Molecular Probes, Eugene, OR, USA)-conjugated antimouse IgG diluted 1:10,000 with 5% BSA in PBS and incubated for 1 h RT. The slides were washed three times with deionized water and stained with DAPI (4’,6-diamidino-2-phenylindole, dihydrochloride) 2 μg/mL (Sigma-Aldrich, St. Louis, MO, USA) for 10 min RT.

The images were acquired in a fluorescence microscope (DMI6000B/AF6000, Leica) coupled to a digital camera system (DFC 365FX, Leica) and processed by the Leica Application Suite X (LAS X). The equipment was granted by the São Paulo Research Foundation (FAPESP), grant number 2012/24105-3.

### 2.6. Carboxyfluorescein Diacetate Succinimidyl Ester (CFSE)-Based Proliferation Assay

CFSE-based proliferation assay was performed as previously described [6]. Mice splenocytes (50 × 10^6^ cells) were labeled for 10 min with 1.25 mM CFSE (Invitrogen, Life Technologies Corporation USA Inc., Waltham, MA, USA) in PBS (37 °C) before being washed with RPMI 1640 medium (Gibco/ThermoFisher Scientific, Waltham, MA, USA). CFSE-labeled cells (3 × 10^5^ cells/well) were expanded with antigen-specific recombinant homologous protein (10 μg/mL), PvCSP variants (10 μg/mL), a pool of synthetic peptides of 15-mer (overlapping by 10 amino acids) spanning the entire sequence of the N-terminal (13 peptides) and C-terminal (13 peptides) of PvCSP (1 μg/mL of each, GenScript, Piscataway, NJ, USA) [6], and mitogen-concanavalin A (2.5 μg/mL, Sigma-Aldrich, St. Louis, MO, USA) in a 96-well plate with “U” bottom (binding Costar 3799, Corning, New York, USA) for five days at 37 °C in 5% CO_2_. On day 5 of in vitro stimulation, expanded cells were washed, collected by centrifugation (300 G, 4 °C), and marked with antibodies/fluorochromes (anti-CD3/APCCy7, anti-CD4/PerCP-Cy5.5, and anti-CD8/PE-Cy7; BD Biosciences, New Jersey, USA) diluted in MACS buffer (BSA (0.5%) (m/v), EDTA (Ethylenediamine tetraacetic acid) (2 mM) PBS, (pH = 7.4)).

Flow cytometric acquisition was performed after a wash with MACS using a 4-color FACSCanto II (BD, Biosciences, NJ, USA) instrument, and analyses were done using the FlowJo^®^ (version 9.0.6, Tree Star) software. A minimum of 200,000 CD3^+^CD4^+^ and CD3^+^CD8^+^ events were acquired. CD4^+^ and CD8^+^ T-cells were gated from lymphocyte population gates based on forward and side scatter. The proliferating cells were identified as populations with decreased mean fluorescence intensity and labeled as CFSE^low^. Results are expressed as the percentage of proliferating cells (groups immunized with the recombinant proteins and restimulated were subtracted from the control groups).

### 2.7. Parasites, Mice, and Mosquitoes

Sporozoites (SPZs) from *Plasmodium berghei* ANKA expressing *P. vivax* CSP-VK210 repeats (Pb/PvCSP-VK210) were obtained as previously described [45]. C57BL/6JRj mice were purchased from Janvier Labs. All animal experiments were approved by the Animal Care and Use Committee of Institut Pasteur (CETEA Institut Pasteur 2013-0093, Ministère de l’Enseignement Supérieur et de la Recherche MESR 01324) and were performed in accordance with European guidelines and regulations (directive 2010/63/EU).

For all tests, female mice aged 6–8 weeks were used and randomly allocated to cages. Two independent immunization/blind challenge experiments were performed using seven animals per experiment as described previously [46]. *Anopheles stephensi* mosquitoes (SDA500 strain) were reared at the Centre for Production and Infection of *Anopheles* (CEPIA) at the Institut Pasteur using standard procedures. For the production of rodent *Plasmodium* spp. SPZs, mosquitoes were fed on infected RjOrl:SWISS mice 1–2 days after emergence and kept in a humidified chamber at 21 °C. One week after infection, Pb/PvCSP-VK210-infected mosquitoes were fed on naïve RjOrl:SWISS mice. For footpad injections, Pb/PvCSP-VK210 SPZs were collected from mosquito-infected salivary glands 21–28 days after the infectious blood meal [46].

### 2.8. Transgenic P. berghei Sporozoite Challenge

Transgenic Pb/PvCSP-VK210 SPZs were maintained in female *A. stephensi* mosquitoes. The total number of SPZs was determined using a Kova glass slide, and 5000 SPZS/μL of PBS was microinjected in the footpad skin using a 35–36 g needle with a NanoFil syringe (World Precision Instruments, Sarasota, FL, USA) in naïve, control, and immunized mice. Parasitemia was determined by flow cytometry performed during days 4–10 after the SPZ challenge. For this, 200,000 erythrocytes were examined for each sample. A quantitative analysis of protection was performed using the parasitemia log values on day 5 postinfection, when the blood parasites were still exponentially growing [46].

### 2.9. Statistical Analysis

The experiments were conducted completely at random, and all the data were tested for normal distribution (Shapiro–Wilk). One-way ANOVA was used to compare normally distributed log-transformed means for the different animal groups. Multiple comparisons were assessed by Tukey’s test, with a *p*-value of 0.05 considered significant.

## 3. Results

### 3.1. Antibody Responses Induced by the Vaccine Formulations in BALB/c and C57BL/6 Mice

In an attempt to study the anti-PvCSP and anti-PvAMA-1 antibody responses elicited when the different formulations were administered to mice, two different strains were used throughout this study: BALB/c (H-2^d^) and C57BL/6 (H-2^b^). Mice were immunized with each recombinant protein (PvCSP-All_FL_ or PvAMA-1) or with the protein mixture ((PvCSP-All_FL_ plus PvAMA-1, (Mix)) using poly(I:C) as an adjuvant. Each animal received three doses 15 days apart, and the antibody titers against each protein were measured by ELISA after two weeks.

In BALB/c mice, the antibody response against PvCSP-All_FL_ was modest (<10^4^ after three doses) and detectable only after two doses. Moreover, the IgG titers were impaired with the combination of antigens (*p* < 0.01). In contrast, the seroconversion to PvAMA-1 occurred after only one dose, reaching >10^4^ after the third dose. We found that the antibody titers generated against PvAMA-1 were similar among groups of mice injected with PvAMA-1 or protein mixture (Figure 2A). On the other hand, in C57BL/6 mice, we observed antibody titers higher than 10^5^ to PvCSP-All_FL_ after three doses. Titers of antibodies to PvAMA-1 were in the range of 10^5^. No statistically significant difference was observed in the serum IgG titers between the groups of mice vaccinated with each protein individually or mixed (Figure 2B).

To evaluate the specificity of anti-PvCSP-All_FL_ antibodies, mice sera were tested against the three FliC-PvCSP repeats (FliC-PvCSP-VK210, FliC-PvCSP-VK247, and FliC-PvCSP-*P. vivax*-like), which contain only the repeats fused to the flagellin (FliC) of *Salmonella enterica* serovar Typhimurium [42]. As a result of these ELISA, we identified a significant predominance of PvCSP-VK210 and PvCSP-*P. vivax*-like repeats compared to the PvCSP-VK247 (Figure 2C, *p* < 0.001), confirming the previous data of our group [7,8].

To better characterize the anti-PvCSP and anti-PvAMA-1 responses, the IgG subclasses were analyzed in both mouse strains, and the IgG1/IgG2a (BALB/c) and IgG1/IgG2c (C57BL/6) ratios were calculated (Figure 3). As shown in Figure 3A, high levels of IgG1 were observed in BALB/c mice, indicating Th2 polarization (IgG1/IgG2a > 1), although the differences between IgG1, IgG2a, IgG2b, and IgG3 were not statistically significant (*p* > 0.05). On the other hand, in C57BL/6 mice, the groups immunized with PvCSP-All_FL_ or protein mixture showed a more pronounced polarization to Th2 response (IgG1/IgG2c>1) (Figure 3B). A comparison of results of the administration of the chimeric protein alone with those obtained in combination revealed that the addition of PvAMA-1 improved the balance of the induced immune response by reducing the differences between IgG1 and the other subclasses evaluated.

The IgG antibody titers generated against the recombinant protein PvCSP-All_FL_ remained unchanged throughout 180 days after the first dose of vaccine formulations (10^4.96^) and those generated against PvAMA-1 for 60 days (10^4.61^). After the decay observed at day 180, the antibody titers against the chimeric protein remained stable for over 63 days, when it suddenly decayed. As for the PvAMA-1 ectodomain, the titers remained in decline (*p* < 0.001) during the entire follow-up of 420 days (Figure 4).

### 3.2. Cell-Mediated Immune Response

To determine whether the vaccination elicited T-cell-mediated immune responses, we evaluated cell proliferation in immunized mice compared to the control. Spleen cells from immunized C57BL/6 mice were stimulated in vitro with recombinant proteins (PvCSP-All_FL_, PvAMA-1, or PvCSP variants) or an overlapping synthetic peptide (15-mer) pool covering the entire length of the PvCSP. The proliferative cell index was determined using CFSE assay.

The results showed the specific proliferation of CD4^+^ T-cells with homologous stimuli in the groups immunized with PvAMA-1 or PvCSP-All_FL_ separately (Figure 5A). Moreover, the pattern of responses to PvCSP-All_FL_ and PvCSP variants observed in the group immunized with PvCSP-All_FL_ was in agreement with serum titers. The same pattern of responses was found in the analysis of CD8^+^ T-cell proliferation in response to these stimuli (Figure 5B). In both cases, we did not detect significant cell proliferation when splenocytes were stimulated with an overlapping synthetic peptide pool.

Interestingly, compared to mice immunized with the proteins separately, the group immunized with Mix showed a slight but significant reduction of both CD4^+^ and CD8^+^ T-cell proliferation in response to PvAMA-1 stimulus. In addition, CD4^+^ responses to PvCSP proteins were absent in these mice, whereas CD8^+^ T-cell proliferation was maintained. Moreover, we were able to detect specific CD8^+^ T-cell proliferation in response to overlapping synthetic peptides in mice immunized with Mix.

### 3.3. Recognition of the Native Protein PvCSP

In addition, we determined whether sera from C57BL/6 mice immunized with formulations containing the recombinant proteins in combination or each one separately reacted to native proteins (SPZs or mature schizonts) of *P. vivax* in immunofluorescence assays. We observed that sera from the groups immunized with PvCSP-All_FL_ or Mix reacted to SPZs of the *P. vivax* CSP-VK210 strain (Figure 6C,E). We also confirmed that sera from groups of mice immunized with PvAMA-1 protein and the protein mixture reacted to mature schizonts (Figure 6D,F). Antibody recognition was specific as control sera from mice immunized with the adjuvant poly(I:C) did not react (Figure 6A,B). Positive controls performed in parallel with anti-CSP-VK210 or anti-PvAMA-1 mAbs were consistently successful (Figure 6G,H).

### 3.4. Pb/PvCSP-VK210 Challenge

Groups of seven C57BL/6JRj mice aged 6–8 weeks were immunized in three doses with an interval of 14 days. Thirty days after the last dose, groups were challenged with 5000 SPZs from *P. berghei* ANKA expressing *P. vivax* CSP-VK210 repeats (Pb/PvCSP-VK210) (Figure 7A).

Parasitemia was evaluated on days 4, 5, and 6 after the challenge, when parasites were exponentially growing in the blood. Linear regression of log parasitemia data showed that the slope of curves did not statistically differ between groups (poly(I:C): 0.82 ± 0.08, PvCSP: 0.88 ± 0.17, PvAMA-1: 0.82 ± 0.09, and Mix: 1.00 ± 0.24; Figure 7B); therefore, the parasite growth rate in the blood was not affected by PvCSP or PvAMA-1 immunization.

No difference was observed between the y-intercept values of linear regressions of poly(I:C) and PvAMA-1 curves (poly(I:C): −4.63 ± 0.85 and PvAMA-1: −4.64 ± 0.46; *p* = 1.00)), as well as between intercepts of PvCSP and Mix curves (PvCSP: −5.47 ± 0.85 and Mix: −6.30 ± 1.23; *p* = 0.40), suggesting that PvAMA-1 immunization does not induce cross-protection in this model. However, the y-intercepts of PvCSP and Mix curves were statistically lower than those of poly(I:C) and/or PvAMA-1 groups (*p* < 0.001, Figure 7B and Appendix A), indicating that PvCSP-All_FL_ conferred partial protection against SPZs expressing the PbCSP harboring the PvCSP-VK210 repeats. Immunization using PvCSP-All_FL_ induced approximately 4- to 5-fold decrease in parasitemia as assessed by cytometry at day 5 postchallenge (Figure 7C).

## 4. Discussion

In the search for a formulation able to overcome genetic variability and provide universal coverage, recombinant proteins representing each individual allelic variant or hybrids containing two or three allelic variants of PvCSP in a single molecule and viral vectors have been tested [6,7,8,18,19,20,21,47]. We and other authors have clearly demonstrated that it is possible to elicit partial protection against *P. vivax* by the immunization of mice with these chimeric recombinant proteins and challenged with transgenic *P. berghei* parasites [7,8,20,48]. In the present study, we assessed the immunogenicity in mice of vaccine formulations consisting of a mixture of antigens, including PvCSP expressed in the pre-erythrocytic stage and PvAMA-1, which is expressed mainly in the erythrocytic asexual stage but is also found in the pre-erythrocytic stage [49].

The chimeric protein PvCSP-All_FL_ [8], which contains immunodominant B-cell epitopes of the central region (repeats) of the three allelic variants VK210, VK247, and *P. vivax*-like fused and the PvAMA-1 were successfully expressed in the yeast *P. pastoris* as previously described [7,39]. Mice of two different genetic backgrounds (C57BL/6 and BALB/c) were used for immunization with the proteins in the presence of the adjuvant poly(I:C), a TLR3 agonist. The proinflammatory Th1-like environment that this adjuvant promotes is believed to contribute to protection in mice against challenge with transgenic *P. berghei* expressing the repeat region of *P. falciparum* CSP [50]. In addition, in natural infections with *P. falciparum*, CD4^+^ T-cell responses have been shown to correlate with protection [51]. Regarding *P. vivax*, formulations containing poly(I:C) as adjuvant were able to elicit protective antigen-specific immune responses using a variety of recombinant proteins [7,8,52,53]. Specifically, antibodies induced by immunization with the protein PvCSP-All_CT_ in poly(I:C) efficiently conferred sterile protection in 4/6 animals challenged (s.c.) with Pb/Pv sporozoites [7]. Here, we selected the full-length PvCSP for immunization and (s.c.) challenge test, as recent data suggest vaccines against *Plasmodium* should include the CSP N-terminal region [28,30]. In addition, humans immunized with long synthetic peptides representing the N- and C-terminus of PvCSP generated antibodies that can inhibit sporozoite invasion in vitro [17].

The antibody response to PvCSP-All_FL_ proved to be dependent on the mice strain, with the observation of high titers of IgG antibodies (10^6^) in C57BL/6, which remained high for up to six months after the last dose. Anti-PvCSP-All_FL_ antibodies, predominantly IgG1, were able to recognize proteins representing the three allelic variants and specifically the repeated regions. This is important as high titers of antibodies against the central repeat region of *P. falciparum* CSP have been recognized as a key factor in conferring protection against malaria [54]. In general, the coadministration of PvCSP-All_FL_ and PvAMA-1 antigens did not compromise the individual antibody′s response in C57BL/6 mice. In contrast, a significant reduction in the IgG titers against PvCSP-All_FL_ was observed in BALB/c mice immunized with Mix. This apparent antigenic competition could be the result of the immunodominance of AMA-1 epitopes presented through strain-specific MHC (major histocompatibility complex) class II in a hierarchical selection phenomenon [55].

The polarized type Th2 antibody profile changed to a more balanced Th1/Th2 profile by the addition of PvAMA-1 to the formulation, indicating that the antigen, rather than the adjuvant, affects the predominant IgG subclass that is produced. It is worth noting that IgG subclass distribution is a useful tool to predict the type of cellular response that a vaccine will probably elicit, but other factors may influence the cytokine profile that is actually obtained. Moreover, although it is known that clinical immunity to *P. vivax* malaria probably requires high levels of cytophilic antibodies against pre-erythrocytic stages and conjunct activation of CD4^+^ T-, B-, and NK cells specific for erythrocytic antigens, the actual success for vaccines that elicit this type of immune response has been limited in the field [56].

Using our vaccination protocol, we detected the cell-specific proliferative responses of CD3^+^CD4^+^ or CD3^+^CD8^+^ T-cells after stimulation with PvCSP-All_FL_ at lower levels, consistent with data from our previous study using the same stimuli but using a bacterial recombinant protein to vaccination [6]. The proliferation (8.3% for CD3^+^CD4^+^ and 3.3% for CD3^+^CD8^+^ T-cells) and pattern of the secretion of the cytokines IFN-γ, IL-2, TNF-α, and IL-10 (data not shown) associated with PvAMA-1 were reduced during the coadministration (6.3% for CD3^+^CD4^+^ and 2.1% for CD3^+^CD8^+^ T-cells). In a comparison of CD3^+^CD4^+^ proliferation associated with PvCSP-All_FL_, the effect of coadministration was even more pronounced: the rate dropped from approximately 4% to not detectable. In the same manner as that explained for the BALB/c strain, antigenic interference could be established when two different epitopes compete for the same HLA (human leukocyte antigen) molecule [55].

Immunofluorescence analyses using sporozoites from the *P. vivax* strain CSP-VK210 and blood-stage isolates demonstrated that these vaccine-elicited antibodies can recognize the native proteins. To evaluate the ability of these antibodies to protect mice, C57BL/6 animals were challenged with transgenic Pb/PvCSP-VK210 sporozoites. Immunization using PvCSP-All_FL_ or Mix induced approximately 4- to 5-fold decrease in parasitemia as assessed by cytometry at day 5 postchallenge, which was not enough to neutralize the Pb/PvCSP-VK210 sporozoite infection. This apparent failure of the full-length PvCSP to provide sterile protection is consistent with a recent study published by Atcheson and Reyes-Sandoval (2020), which demonstrated that a truncated form of PvCSP missing the N-terminal region was able to confer higher levels of protective efficacy than full-length PvCSP [52]. The precise reason for this difference is not clear. One possibility is that perhaps the presence of two subdominant B epitopes in the N- and C-terminal regions flanking the repeats of PvCSP can mask the immune recognition targeting immunodominant B epitopes in the repetitive central region by altering the hierarchy of epitope recognition.

Importantly, the RTS,S vaccine does not include the N-terminal region of the *P. falciparum* CSP, which contains an important linear epitope [30]. Moreover, our previously studied recombinant PvCSP-All_CT_ recently underwent preclinical safety assessment and was demonstrated to possess all the requirements necessary to advance into clinical evaluation (unpublished data). Therefore, we will now proceed to manufacturing and clinical assessment under good manufacturing practice (GMP) guidelines. Nonetheless, we believe that comparative studies such as this are necessary to exploit the potential of multiallelic, multiepitope, and multistage vaccine formulations.

Based on that, we consider that the data generated on the study of vaccine formulations presented in this work may be useful for the development of an anti-*P. vivax* vaccine, mainly because we explored strategies for the fusion and combination of antigens from more than one stage of the parasite’s life cycle.

## Figures and Tables

**Figure 1 microorganisms-08-00916-f001:**
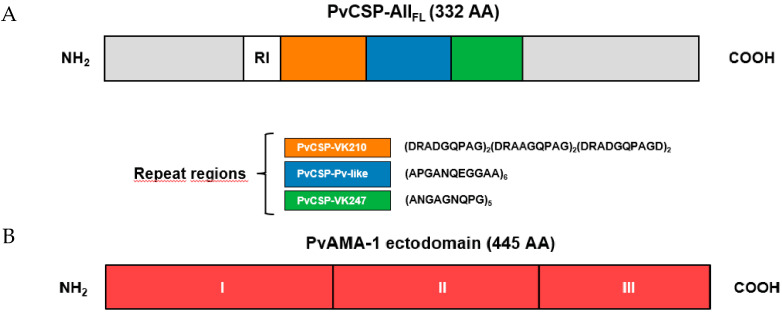
Representation of *Plasmodium vivax* proteins expressed in *Pichia pastoris*. Schematic representation of the PvCSP-All_FL_ construction. Sequences from variant repeats in the central region are indicated (**A**). Schematic representation of PvAMA-1 ectodomain and its subdomains (DI, DII, and DIII) (**B**).

**Figure 2 microorganisms-08-00916-f002:**
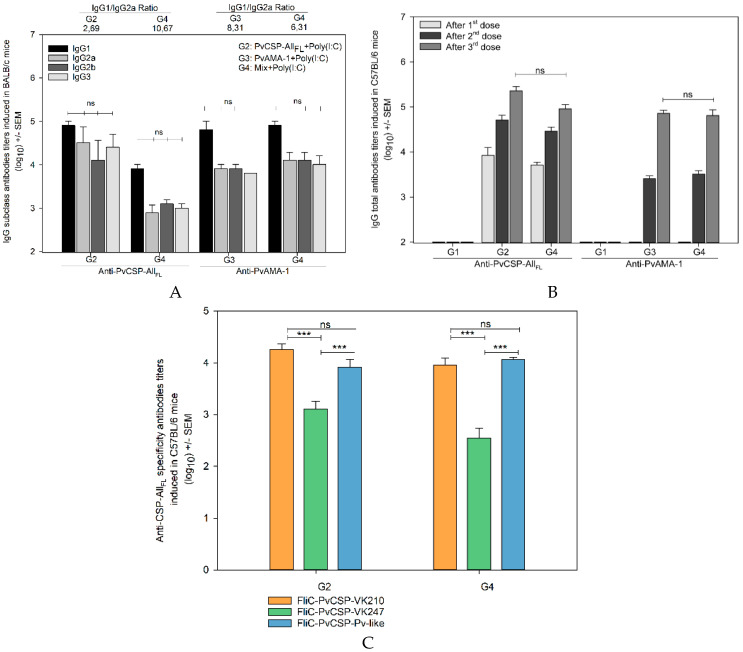
IgG antibody response in mice immunized with the formulations containing PvCSP-All_FL_, PvAMA-1, and poly(I:C) adjuvant. Groups of female BALB/c (**A**) and C57BL/6 (**B**) mice aged 6–8 weeks were immunized (s.c.) with 10 µg of each recombinant protein and poly(I:C) (50 µg/dose). The IgG titers were measured by ELISA against PvCSP-All_FL_ and PvAMA-1. The antibodies anti-PvCSP-All_FL_ recognized the different allelic variants of PvCSP and were directed to central repeats. The anti-PvCSP-All_FL_ antibodies, induced in C57BL/6, were tested for their recognition of recombinant proteins representing the PvCSP variants (VK210, VK247, and *P. vivax*-like), and these repeats were fused to flagelina FliC (FliC-PvCSP repeats) (**C**). The results are expressed as the arithmetic mean titers of each group in log_10_ ± SEM and were statistically compared using one-way ANOVA followed by Tukey‘s test for multiple comparisons. Significant differences between groups are denoted on the graph: *** *p* < 0.001. Nonsignificant (ns) differences are indicated (*p* > 0.05).

**Figure 3 microorganisms-08-00916-f003:**
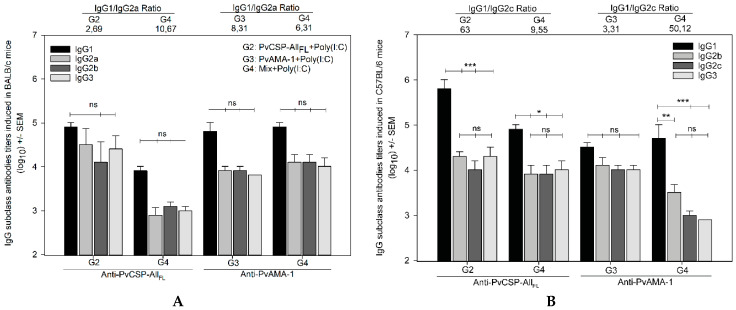
IgG antibodies subclass titer analysis. The IgG antibodies subclasses IgG1, IgG2a, IgG2b, IgG2c, and IgG3 induced in BALB/c (**A**) and C57BL/6 (**B**) mice were measured by ELISA against PvCSP-All_FL_ and PvAMA-1. The results are expressed as the arithmetic mean titers of each group in log_10_ ± SEM and were statistically compared using one-way ANOVA followed by Tukey’s test for multiple comparisons. Significant differences between groups are denoted on the graph: * *p* < 0.05, ** *p* < 0.01, and *** *p* < 0.001. Nonsignificant (ns) differences are indicated (*p* > 0.05). The titers were used to calculate the IgG1/IgG2a (**A**) and IgG1/IgG2c (**B**) ratio, and the ratios are indicated above the plot according to the groups.

**Figure 4 microorganisms-08-00916-f004:**
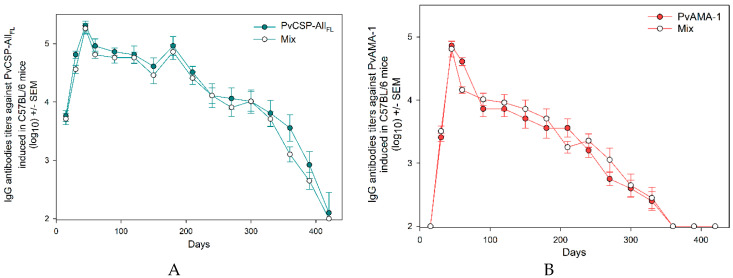
IgG antibodies longevity analysis. The IgG antibody longevity induced in C57BL/6 mice was monitored for 420 days after the first vaccine dose by ELISA against PvCSP-All_FL_ (**A**) and PvAMA-1 (**B**). The results are expressed as the arithmetic mean titers of each group in log_10_ ± SEM.

**Figure 5 microorganisms-08-00916-f005:**
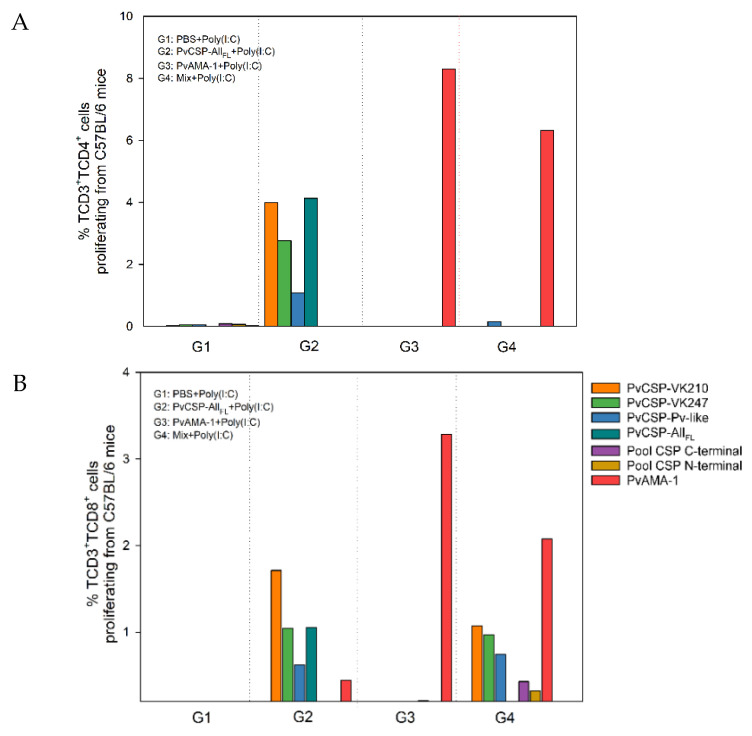
Lymphocyte proliferation from C57BL/6 mice immunized with formulation containing PvCSP-All_FL_, PvAMA-1, and poly(I:C) adjuvant. Pooled splenocytes were collected from immunized C57BL/6 mice. The cells stained with carboxyfluorescein diacetate succinimidyl ester (CFSE) were plated and stimulated (for five days) with homologous proteins or one of the PvCSP variants (10 μg) or pooled synthetic peptides (26 μg), which covered the entire length of the C- or N-terminal PvCSP or mitogen-concanavalin A (2.5 μg). The events were acquired in the FACSCanto II and analyzed using the software FlowJo^®^. The results are expressed as percentage (%) of TCD4^+^ (**A**) or TCD8^+^ (**B**) proliferative cells. These data are representative of two independent assays from a pool of three immunized mice.

**Figure 6 microorganisms-08-00916-f006:**
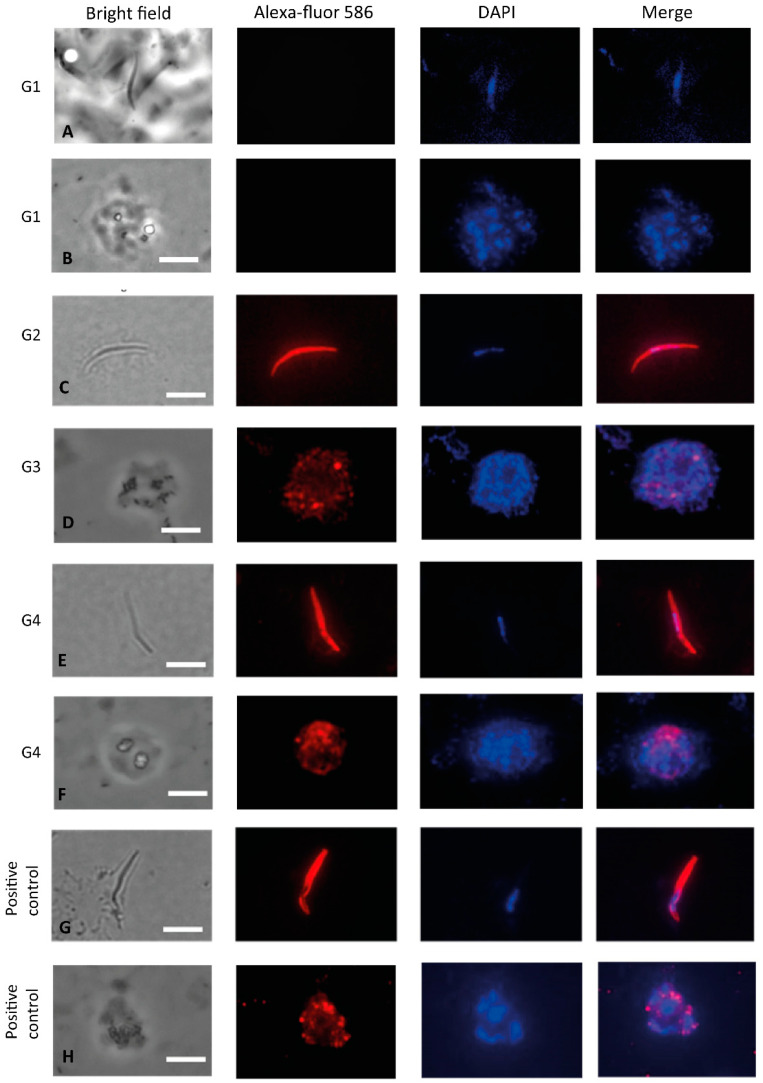
Indirect immunofluorescence analysis using sera from C57BL/6 mice. Microscope slides containing fixed sporozoites of *P. vivax* obtained from patients from Thailand (**A**,**C**,**E**,**G**) and merozoites of *P. vivax* obtained from patients from Rondônia (Brazil) (**B**,**D**,**F**,**H**) were incubated with a pool of sera from mice immunized with PvCSP-All_FL,_ PvAMA-1 or protein mixture in the presence of poly(I:C). (**A**,**B**) G1: negative control sera from PBS + poly(I:C), (**C**) G2: polyclonal sera anti-PvCSP-All_FL_ (1:100), (**D**) G3: polyclonal sera anti-PvAMA-1 (1:100), (**E**) G4: polyclonal sera anti-Mix (1:100), (**F**) G4: polyclonal sera anti-Mix (1:100), (**G**) positive control mAb anti-CSP-VK210 (1:100), and (**H)** positive control mAb K243 anti-PvAMA-1 (1:1000). The white bars are equivalent to 10 µm.

**Figure 7 microorganisms-08-00916-f007:**
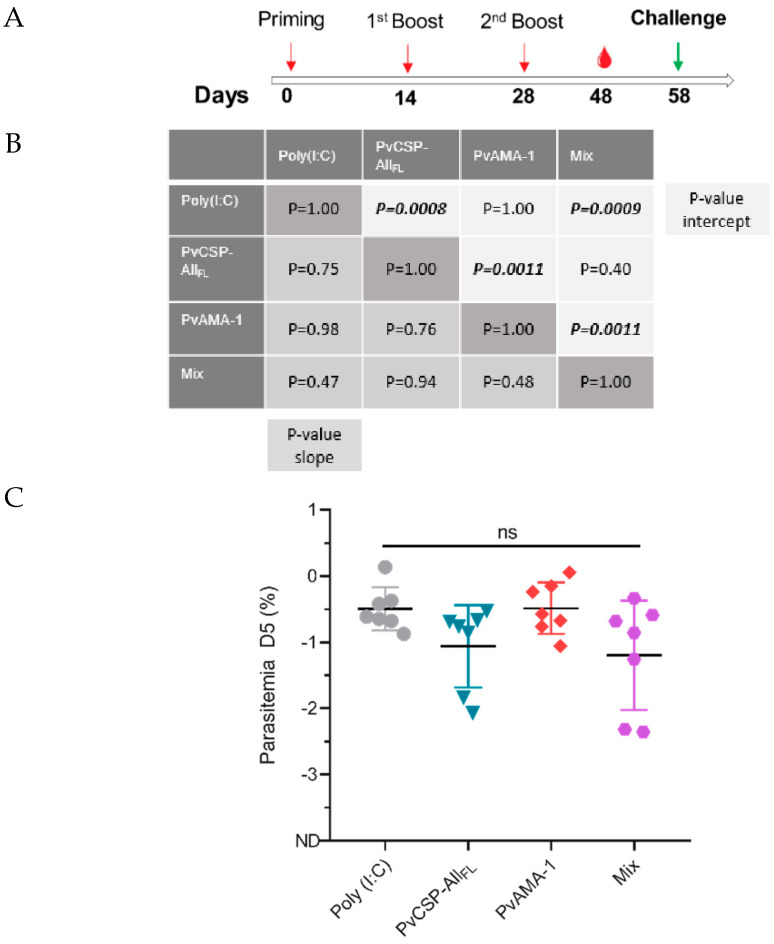
Evaluation of parasitemia after challenge in immunized mice. Groups of female C57BL/6 mice aged 6–8 weeks (n = 7) were immunized (s.c.) with 10 µg of each recombinant protein and poly(I:C) (50 µg/dose). On day 58 after priming, the mice were challenged with 5000 Pb/PvCSP-VK210 transgenic sporozoites (**A**). Parasitemia was analyzed by flow cytometry. The percentage of infected red blood cells (iRBCs) on days 4, 5, and 6 postinfection (p.i.) was expressed as log values for normalization before statistical analysis (**B**). The log of parasitemia on day 5 (D5) postchallenge was measured in mice in each of the immunized groups **(C**). Data from two independent experiments and significance were determined by two-tailed unpaired T-test (Mann–Whitney test).

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
