# Peer review of "A Multistage Formulation Based on Full-Length CSP and AMA-1 Ectodomain of Plasmodium vivax Induces High Antibody Titers and T-cells and Partially Protects Mice Challenged with a Transgenic Plasmodium berghei Parasite"

_microorganisms, 2020, doi:10.3390/microorganisms8060916_

Round 1

Reviewer 1 Report

Plasmodium vivax is clinically the second most important malaria-causing parasite species. It is endemic in middle and south America as well as large parts of Asia, putting about 2.5 bn people worldwide at risk of infection. Although infections with P. vivax are often considered more benign than those caused by P. falciparum, they can nevertheless severely affect human health. To date, no effective vaccine has been developed which protects against infections with P. vivax parasites.

In this manuscript, Lima and colleagues have investigated the efficacy of two different vaccine candidates in mice, namely P. vivax full-length CSP and the AMA1 ectodomain, administered either alone or in combination.  Although PvCSP and PvAMA1 alone induced high antibody titers in mice which lasted for about 200-300 days, and a low-level CD4+ and CD8+ T-cell proliferation response, they failed to induce immunity against infection with chimeric PvCSP expressing P. berghei parasites. Interestingly, combining both vaccine candidates into a single formulation did not improve the immune response.

Despite this somewhat disappointing outcome, the study appears scientifically sound and will be useful in informing future studies in their attempts to develop an effective vaccine against P. vivax infections in humans.

The following issues should be addressed by the authors:

  1. Overall: The quality of all figures in this manuscript could be improved. The authors should consider adding colour to their plots, add additional labelling and potentially split plots into multiple separate plots. The authors should also indicate which graphs belong to respective panels A, B, C, … .
  2. Section 3.1, Figure 2: Fig. 2C seems missing from the manuscript.
  3. Section 3.1, Figure 3: Fig. 3 is difficult to understand as it tries to display IgG1/IgG2 ratios and IgG subclass titers in a single plot. The authors should consider ways to enhance clarity, potentially presenting IgG1/IgG2 ratios and IgG subclass titers in individual plots.
  4. Section 3.1/3.2: The authors should indicate what level of IgG titers and T-cell response are typically required to induce protection against a pathogen such as malaria (e.g. data from comparable studies on P. vivax or P. falciparum vaccine candidates) and compare them to the results of their study. This will enable the reader to better appreciate the findings of this study and to put them into the wider context.
  5. Section 3.2, Figure 5: The plots in Fig. 5 are difficult to read. The authors should enhance clarity to allow readers easier access to their research findings. Fig. 5 also lacks error bars and information about biological replicates or no. of analysed animals.
  6. Section 3.3, Figure 6: The authors should improve labelling of this figure. The figure lacks an indication of the identity of individual image rows.
  7. Section 3.4: The authors should explain in more detail how the parasite challenge was performed and include information about the parasite used (species, genotype, etc). Although the authors have included this information in Section 2. of the manuscript, repeating this information in the results section would clearly increase the understandability of the respective section.
  8. Section 3.4, Figure 7: It is unclear which images/graphs belong to panel A, B and C in Fig. 7, and what are actually panels A, B and C. The authors should also present the original (not-log transformed) parasitaemia after parasite challenge. Otherwise, it is difficult to see where the reduction in parasitaemia of 4-5-fold after immunisation with PvCSP is coming from. This is especially important, as the authors base their main conclusion on these results.
  9. Section 3.4: The text in section 3.4 is at times confusing, as it is unclear which data and which part of the text refers to which particular panel in Fig. 7. The authors should enhance the readability of this particular section. The authors should also include graphs for each individual challenge as four separate graphs in the manuscript (shown in the current version of Fig. 7 (lower right panel) as four differently coloured groups of data points), including key parameters such as slopes and intercepts used to calculate p-values displayed in Fig. 7 (lower left panel).
  10. Section 3.4, Figure 7: Fig. 7 lacks information about the number of mice analysed.

Reviewer 2 Report

The manuscript presented by Lima et al shows a proof of concept of an experimental immunization process in the context of malaria, specifically caused by Plasmodium vivax. The study is relevant, the results are interesting and are well presented, however it needs some comments/suggestions:

1. In the introduction (page 2 line 55) the authors need to explain what the AMA-1 antigen means;

2. In the introduction and discussion, the authors need to clarify why the use of TLR3 agonist adjuvant. How important is this agonist in the immunization process in the context of malaria?

3. Page 2 (lines 64-82) - I think the paragraph is too long and therefore I suggest that it be restructured;

4. Page 3, line 89 - What does QuilA adjuvant mean? I think it should be better clarified in the manuscript;

5. Why use 10 ug of antigen? Are there any previous studies that optimize this amount? Would not it be too low an antigen to induce protective and lasting immunity?

6. The authors report that the immunized animals mainly present antibodies of type Th2, mainly IgG1 antibodies. How important are antibodies of type Th1 versus Th2 in inducing immunity during natural infection with Plasmodium vivax? Is a polarized Th2 response and IgG1 production an important protective mechanism in the immunopathogenesis of malaria? I think these issues should be raised in the discussion;

7. Page 13 (line 431) - the authors refer to the importance of Th1 cytokines in malaria, specifically IFN-gamma, IL-2, IFN-alpha, however, previously they mentioned the importance of the polarized response of antibodies of type Th2 (IgG1). Therefore, I think that the authors should better explain the importance of the polarized response (Th1 versus Th2) in the context of the natural infection and malaria vaccination process.
